# The Real Deal: A Qualitative Investigation of Authentic Leadership in Irish Primary School Leaders

Jemma Lynch, Dympna Daly, Niamh Lafferty * and Patricia Mannix McNamara

School of Education, University of Limerick, V94 PX58 Limerick, Ireland; Jemma.lynch@ul.ie (J.L.); Dympna.Daly@ul.ie (D.D.); Patricia.M.McNamara@ul.ie (P.M.M.)
* Correspondence: Niamh.lafferty@ul.ie

**Abstract:** Recognition of the importance of authentic leadership is growing in popularity amonleadership scholars. However, little remains known about how it is valued or received among practicing school leaders. The purpose of this research was to explore the perspectives and experiences of school leaders with reference to authentic leadership in Irish primary school leaders. As this is a scoping study, a qualitative research design was adopted, using semi-structured interviews with school leaders. Core traits of self-awareness, balanced processing, relational transparency and internalized perspectives, that are associated with authentic leadership emerged as important for those interviewed. Barriers and facilitators of authentic leadership were also identified including educational policy, procedures and school culture.

**Keywords:** authenticity; authentic leadership; educational leadership; leadership; primary schools





## 1. Introduction

As with leaders in any organization, school principals play a pivotal role in the overall culture of the schools they lead [1,2]. School leadership has been shown to influence organizational commitment [3], organizational climate [4], organizational change [5] and also has a significant influence on workplace bullying [6]. The influence of school leadership does not stop with employees but also has influence and impact on the wider school community, including students [7–9] and parents [10,11].

Considering the pivotal role school leaders play in school culture, it is unsurprising that educational leadership has received increased research attention in recent years. It has also drawn the attention of pedagogists who recognize the influence of leadership on cultures of teaching and learning. Within the literature, particular emphasis has been placed on leadership styles [12] with a systematic review from Gumus et al. [13] highlighting emergent leadership models such as distributed, transformational and instructional leadership approaches. Outside of educational settings, authentic leadership [14] has been gaining increasing traction, with research emerging from industries such as healthcare [15], service companies [16] and hospitality [17]. This increased focus has been linked, in part, to the potential benefits of authentic leadership in positively influencing employee performance and organizational outcomes [14,15]. Furthermore, it has been identified as a possible solution to tackling unethical behaviors from corporate leaders [18,19]. Authentic leadership has even been referred to as the "gold standard of leadership" [20] (p. 54).

Despite these potentially positive outcomes, there is relatively little representation of authentic leadership research in educational leadership literature. This is highlighted in the limited reference to, and often complete absence in numerous systematic and meta-analytical reviews of, leadership styles in education [13,21] and leadership style scales [22–24]. Although research has been carried out in numerous countries, the majority of existing literature on authentic leadership has emerged from the US and Canada which was initially highlighted by Gardner and colleagues in their 2011 systematic review and by

more recent literature reviews such as Alliyani and colleagues in 2018 [15] and Liebowitz and Porter in 2019 [25]. Notably, an Irish context is relatively absent from the literature and, given the impact of national culture on work values [26] and leadership styles [27], this gap can only be filled by carrying out research in the country of Ireland itself. The research described in this paper will directly work toward broaching this gap.

### 1.1. Authentic Leadership Definition

Core assumptions underpinning authentic leadership are rooted in philosophical underpinnings of the true self [28,29]. Authentic Leadership as a construct was first formally conceptualized by Terry [30] in his seminal work Authentic Leadership: Courage in Action. Terry's development of the Zone Leadership Model [30] was the first exploration into examining the traits of authentic leadership and how these are evident in organizations. Terry proposed that the importance of the individual cannot be underestimated and advocated the establishment of leadership frameworks "based on each person's own definition of leadership" [30] (p. 2).

The concept of authentic leadership suggests that authentic leaders' actions are based on their own moral and ethical beliefs [19,30]. As leaders' actions are guided by and in alignment with their own values and ethics, this has been expected to promote trust, authenticity, well-being and sustained performance in their followers [28,31]. It should be noted here that these values and ethical beliefs are often assumed to be socially normative [32] however, this may not always be the case [33] and one's values may in fact be perceived as unethical by social standards. This will be discussed in more detail later in the paper when discussing "internalized moral perspective".

Despite the widely accepted fundamental characteristics of authentic leadership, inconsistency remains across the literature in terms of an explicit definition. To tackle this discrepancy, Cooper, Scandura and Schriesheim [34] proposed the need for a theoretically based definition of authentic leadership through the identification of key dimensions. In attempting to identify these key dimensions, Ferrero et al. [35] then proposed a framework based on Thomas Aquinas' virtue of ethical wisdom [36], encapsulating four characteristics, i.e., self-awareness, relational transparency, balanced processing and internalized moral perspective (Table 1).

**Table 1.** The characteristics of the authentic leader according to Ferrero et al. [35] (p. 88). Reprinted with permission from Ref. [35]. 2020, John Wiley and Sons.

| | |
|---|---|
| Self-awareness | The leader's relationship with themself |
| Relational transparency | The leader's relationship with others |
| Balanced processing | The leader's relationship with organizational processes and decisions |
| Internalized moral perspective | The leader's relationship with the values he or she holds |

These four characteristics provide a strong basis from which to conceptualize authentic leadership. Ferrero et al. [35] explain these concepts in the following way.

### 1.1.1. Self-Awareness

Self-awareness is achieved through reflective thinking, during which a leader reflects on their "values, goals, knowledge, beliefs, sense of purpose, talents, strengths, and weaknesses" [35] (p. 88). This type of reflection should positively influence balanced decision making. Due to this, authentic leaders would be expected to spend time reflecting on their past experiences, their personal values/morals and assessing their own competencies, strengths and weaknesses [31,37].

### 1.1.2. Relational Transparency

Relational transparency requires basing one's actions as a leader on transparency and is driven by integrity and strong communicative competence characterized by clarity. Relational leaders represent themselves honestly, through clear and direct communication of their personal values and goals.

### 1.1.3. Balanced Processing

Balanced processing requires thoughtful interrogation of both the organization's priorities and the needs of its teams. Decisions are not made impulsively and feedback plays a key role in influencing decision making. Balanced processing can minimize biased thinking [38] by allowing for the consideration of others' perspectives. For balanced processing to occur, leaders will seek out the opinions of others [37], listen to the ideas of those who disagree with them [37] and take time to reflect upon the information they receive from others [37].

### 1.1.4. Internalized Moral Perspective

An internalized moral perspective identifies the importance of a moral compass for authentic leaders whose actions are guided by ethical principles. Ferrero et al. [35] (p. 88) identify that it is this that sets authentic leadership apart from "inspirational, charismatic, or even narcissistic types of leadership" [35] (p. 88). In respect of this, authentic leaders will have a good understanding of their personal values and morals, and this will guide their decision-making behaviors. At first glance, these may seem easy to embody, however, authentic leaders who seek to ensure their behavior is congruent with their values, may then be met with organizational cultures, policies and practices that may be at odds with these values. Then being able to be one's authentic self may be harder to achieve. This resonates with what Jack Whitehead [39] calls a 'living contradiction,' which he explains as the experience of holding educational values and the experience of their negation at the same time.

It is important at this stage to discuss the elements of values and morals at the core of authentic leadership for the purpose of clarity. When discussing high morals and being authentic to one's values, it is the leader's beliefs of what is moral that are in question [40,41], not a societal or otherwise external embodiment of moralistic ideals. There is a differentiation here between behaviors that may be seen by others as un/ethical or a/moral, and the personal values held by the leader that guides them to behave in such a way. One's personal belief about what is ethically correct, and what they believe is the "right thing to do", may be at odds with popular opinions and even common sense, however, it is the commitment to their beliefs and the extent to which the leaders' behavior is influenced by their beliefs, that is at the core of authentic leadership, i.e., they are being authentic to their personal values and sense of morality [40,41].

Ethical values and morals may differ between people and leaders are no different in this regard. With that said, a leader's values may align with perceivably unethical behavior, in which case, a truly authentic leader may behave unethically. However, the scope and focus of this research are not to ascertain what constitutes moral behavior, only to explore whether school leaders are leading based on their morals and values and in so doing, exhibiting traits of authentic leadership.

### 1.2. Authentic Leadership: A Distinct Style

As discussed previously, authentic leadership offers an alternative to more traditional leadership styles which have been associated with unethical behavior and detrimental outcomes for multiple stakeholders. Identified as one of the three ethical/moral values-based leadership forms [40] authentic leadership stands with ethical and servant leadership as a potential solution to opposing leadership styles which may be deemed unethical and detrimental to multiple stakeholders [40]. Although leaning on the theory of transformational leadership, authentic leadership differs through the emphasis on one's core

ethical and moral beliefs [30]. Whereas the distinction between authentic leadership and transformational leadership can be quite easily distinguished, the overlap between ethical leadership and servant leadership is slightly more cumbersome and unclear in much of the existing literature. To allow for clarity, a distinction between these styles is necessary.

Firstly, in comparison to servant leadership, authentic leadership differs due to the leader focus. For servant leadership, the primary focus is very much on the followers, with the aim being to serve their followers [42] as opposed to the organization [40,43]. For the authentic leader, however, both the organization and the followers are taken into consideration [37].

In relation to ethical leadership, there are many similarities with authentic leadership, such as the concern for others, ethical decision making, integrity and role modelling [44]. The key differences are, in the first instance, that authenticity specifically includes considerations for self-awareness whereas ethical leaders focus on other-awareness [44,45]. Secondly, ethical leadership lacks the focus on authenticity specifically [45] and includes transactional leadership behaviors which authentic leadership does not [40,44].

An additional form of leadership to be considered when discussing leadership in a school setting is distributed leadership. Distributed leadership has been strongly endorsed by the Organization for Economic Co-Operation and Development (OECD) as a means of accommodating increased numbers of duties placed on academic staff [46]. It is now widely present both in practice and policy [47–49].

In contrast to authentic leadership, distributed leadership can be described as a social practice rather than an individualistic leadership style [50,51]. Despite some suggestions that distributed leadership may have benefits for teachers [52], the organization [53,54] and the students [55–57], this leadership approach is arguably becoming a product of the neoliberal agenda in which accountability and performativity standards are heavily increasing resulting in increased workloads and demands on teachers [46,48,58].

To successfully implement distributed leadership as an empowering process, as opposed to a burdening of employees [59] requires a culture of trust [52,60] and communication [60], where staff are listened to and involved in the distribution of tasks which match their skills and competences [60]. In respect of this, as opposed to replacing the distributed leadership process in schools, authentic leadership may promote the successful implementation of the process. Authentic leadership has been linked to increased levels of trust [61] and communication through the promotion of employee voice behaviors [62]. Given the dominant role of the distributed leadership process within schools, this adds further merit to the importance of studies pertaining to authentic leadership as it may negate the potential negatives and promote the benefits.

### 1.3. Challenges to Authentic Leadership Enactment

As discussed previously, authentic leadership is rooted in the ideal of being "true to oneself" and one's personal values. In consideration of this facet, despite the potential benefits of an authentic leadership positioning, the ability of one to lead authentically may only be facilitated when there is an alignment between the leader's values and that of the organization. Previous research has identified value-congruence as a key predictor of leadership success when looking at other leadership styles such as charismatic leadership [63], ethical leadership [64] and transformational leadership [65], but to date, there is no research, to the authors' knowledge, in relation to authentic leadership. This is quite a significant gap considering the importance of values at the center of the authentic leadership theory.

Furthermore, the ability to lead based on one's core values may be a freedom awarded to only a select few, with marginalized groups being excluded [32]. At present, the necessary recognition of power dynamics [66] is predominantly absent from literature and research on authentic leadership. Although this is beyond the scope of the current study in terms of marginalized groups' experiences of authentic leadership, it does highlight the necessity for consideration of individual barriers that may prevent some individuals from leading authentically.

*1.4. Antecedents of Authentic Leadership*

Antecedents of authentic leadership have been identified both at the personal and organizational levels. Leader self-knowledge and consistency have been identified in previous research as antecedents to authentic leadership [67]. It is unsurprising then that emotional intelligence has also shown to be positively associated with authentic leadership [68], given that those with strong emotional intelligence have a greater awareness and comprehension of their own emotions [31] which resonates well with self-awareness.

From an organizational perspective, the literature shows that the leadership styles adopted by leaders are influenced by the organization they work in [69]. Organizational ethical climate i.e., "employees' shared perceptions of organizational practices and procedures that help individuals determine what to do when making decisions related to the organization or its members" [68] (p. 3) is also influential [70]. What this appears to suggest is an intersection between the personal and the organizational dimensions in the development of leadership styles, a clear departure from traditional understandings of the charismatic and hero leader.

The literature indicates benefits of authentic leadership approaches both for followers and for the leaders themselves [68]. Leader benefits have been found in relation to a positive influence on leader–member exchange [68], feelings of leader effectiveness [68] and self-esteem [71]. For followers, responses to authentic leadership increase the level of satisfaction with the leader [67], with organizational commitment [37,67,68], for organizational citizenship behavior [38], psychological empowerment [68], psychological safety [68] and workplace trust [68]. As well as improving numerous positive elements for followers, authentic leadership has also been associated with a reduction in many negative aspects, such as emotional exhaustion [68,72,73], turnover intention [68,74] and cynicism [68].

While the literature on authentic leadership in the education setting is more limited, in what does exist, evidence for the benefits of authentic leadership for teachers includes having a positive impact on teachers' psychological empowerment [75] and decreases in emotional exhaustion in followers [76]. Given the current recruitment and staffing crisis being experienced in teaching and in educational leadership globally [75,77–80], focusing on more positive and sustainable leadership approaches is vital. Given the potential benefits of authentic leadership for the employee, particularly in relation to reduced turnover intentions [74], there is strong support for this to be considered in addressing the current crisis.

Despite being deeply rooted in the idea of being 'true to oneself' [81] (pp. 2–3) and dictated by one's personal morals and ethical beliefs [19], research on authentic leadership has often focused on the perspective of the followers [37,82,83]. This approach has been criticized for focusing less on leaders' actual authenticity and more on the subjective assumption of authenticity from the followers' point of view [84–86]. The cause of concern with such an approach is that without exploring the leaders' reality, there is no benchmark with which to measure if their seemingly authentic displays are anything other than convincing acts [87].

Given the paucity of research into authentic leadership in Ireland, this study is an initial scoping study to explore how school leaders themselves conceptualize and practice authentic leadership. Researchers in this field tend to explore this field via quantitative questionnaire design [37,82,88–90]. However, in heeding the calls for different approaches to studying authentic leadership [86] an initial qualitative in-depth study was designed to listen to the perspective of school leaders themselves.

*1.5. Research Purpose*

The purpose of this research is to explore authentic leadership in an Irish, primary school context which to date, is absent from the existing literature. Not only will this work towards filling this substantial gap but will also answer calls for an increase in research on authentic leadership in general [86], an increase in research on authentic leadership incorporating a qualitative design [86] and an increase in research on alternative approaches

to distributed leadership in schools [91]. Given the initial scoping nature of the study, it is not intended to seek to generalize from this study but rather to begin a conversation about authentic leadership and to lay the foundations for future research in this context.

## 2. Materials and Methods

An interpretative qualitative research design was adopted for this study. This comprised semi-structured interviews to explore how a sample of Irish school leaders perceived their leadership approaches and to analyze for the existence of authentic leadership traits of self-awareness, relational transparency, balanced processing and internalized moral perspective [35]. This approach was adopted as the researchers sought to understand experiences of leadership in primary schools with the purpose of deepening understanding of leadership in this particular context. The authors' research ethics committee granted approval for the study.

### 2.1. Instrument Development

Interviews were chosen as the data collection method because they would allow the researcher to explore in more depth how interviewees conceptualized and experienced leadership. Rigorous data collection procedures are essential for ensuring high quality [92], trustworthy [92] and reliable research findings [93]. To ensure rigor at every stage of the research and to avoid having the credibility, validity and reliability of the research being called into question [94], steps outlined by Kallio et al. [95] were followed for the development of a semi-structured interview schedule to address the main research aims.

### 2.1.1. Identification of Semi-Structured Interview as Most Appropriate Method

The purpose of this research was to gather information pertaining to the participants' perceptions and experiences, as well as to explore areas relating to their values which, as discussed in the introduction, are at the core of authentic leadership. Semi-structured interviews have been identified as appropriate for carrying out research on perceptions and opinions [95,96] and values [95,97]. For that reason, semi-structured interviews for this study were deemed appropriate.

### 2.1.2. Retrieval and Implementation of Previous Knowledge

Previous literature and consultation with subject matter experts (research experts, experts in the field of educational leadership and former school principals) guided the development of the interview guide. The critical appraisal of the literature has been previously discussed in the introduction section of this paper. Consultation with research experts allowed for the consideration and implementation of open-ended questions to encourage participant conversation, the identification and eradication of ambiguous or confusing terms and jargon to allow for participant understanding, consideration and inclusion of prompts and probes in the interview schedule and the identification and eradication of potentially offensive items, which may cause harm to the participants or negatively impact agreement to participate. Consultation with topic experts (former principals and educational leadership experts) further allowed for the identification of jargon and ambiguous terms as well as suggestions of questions that would encourage the flow of conversation.

### 2.1.3. Formulation of Interview Schedule

In drawing from the information gathered in the previous step, the interview schedule was drafted. Questions were open-ended, free of jargon or ambiguous terms and directly targeted at addressing the main research aims. To identify authentic leadership traits, items were included to explore the elements of self-awareness, relational transparency, balanced processing and internalized moral perspective. Examples of items are included in Table 2. Participants were also asked more general questions about their leadership journey to the point of interview and examples of times when they felt that they, as leaders,

had had a positive impact in their schools. The latter items were included to improve the benefits of research participation for the individual participants, by providing them with an opportunity to have their voices heard and reflect on positive times in their work careers [98].

**Table 2.** Example of items.

| AL Trait | Aspect(s) of Trait | Sample Interview Item |
|---|---|---|
| Self-Awareness | Personal reflection. Reflection influences decision making. | Throughout your leadership experience, have you been able to reflect on your past experiences? Probes: If not, why? If yes, can you give an example? |
| Relational Transparency | Active-listening Communication Honesty | How receptive are you to the ideas, opinions, and experiences of others? |
| Internalized moral perspective | Awareness of morals. Morals influence decision making. | If you were to outline your own personal morals with regards to educational leadership, what would they be? |
| Balanced-Processing | Evidenced-based decision making Openness to feedback Balancing outcomes for all stakeholders | When making difficult decisions, what are your main considerations? |

### 2.1.4. Pilot Testing

Two pilot studies were conducted which allowed for the refinement of the schedule in terms of question clarity and the elimination of ambiguous terms. Data from the pilot studies were not included in the final data analysis. The pilot testing also allowed for the consideration of other elements of the interview process such as suitability of the location, devices used for recording and length of the interview. These considerations allowed for the ethical and procedural elements of the data collection to be greatly enhanced as location security and anonymity were evaluated and participation time was more accurately estimated.

### 2.2. Participant Demographics

A total of ten participants, six males and four females, participated in the semi-structured interviews (See Table 3). Participants represented a mix of schools including single-sex and mixed-sex student cohorts; Catholic and non-denominational; DEIS (Delivering Equality of Opportunity in Schools, for schools whose socio-economic context would be deemed disadvantaged) and non-DEIS; and Gaelscoil and non-Gaelscoil.

**Table 3.** Participant characteristics.

| Pseudonym | Years Teaching | School Type |
|---|---|---|
| Nuala | 2 | Gaelscoil, Co-ed, Catholic |
| Lucy | 10 | Co-Ed, Catholic, DEIS |
| Shane | 21 | Co-Ed, Catholic |
| Daniel | 22 | Co-Ed, Catholic |
| Rita | 16 | Boys, Catholic |
| Anita | 10 | Co-Ed, Catholic, DEIS |
| George | 11 | Co-Ed, Educate Together |
| Eric | 15 | Co-Ed, Catholic |
| Declan | 12 | Co-Ed, Catholic |
| Steve | 13 | Co-Ed, Educate Together |

Note: Co-ed = Co-Education.

### 2.3. Data Collection

Due to COVID-19 restraints, the semi-structured interviews were conducted online using Microsoft Teams, a GDPR-compliant platform, at a time convenient to the interviewee. Each interview lasted between 30 and 70 min. Audio recordings collected from the interview were saved securely until anonymized transcripts were generated, following which, the audio files were destroyed for confidentiality purposes.

Participants were recruited via convenience sampling. Contacts of the research team who were known to work as primary school leaders were sent an information sheet via email containing details of the study and ethical considerations. This leaflet also contained the contact details of the research team to allow the participant to ask any questions they might have or to agree to participate. Following contact with the research team, if they chose to participate, they were sent a consent form and interview times were scheduled.

### 2.4. Data Analysis

Data were analyzed using Smith, Flowers and Larkin's [99] model of Interpretative Phenomenological Analysis (IPA). This model is an iterative and inducive cycle [100] that is comprised of several sequential steps that were followed closely for this analytical procedure. Following the strategies for completing IPA outlined by Smith et al. [99] (pp. 79–80) the first step required close, line-by-line analysis of the experiential claims, concerns and understandings of each participant. This then led to the identification of emergent patterns (i.e., themes) within this experiential material emphasizing both convergence and divergence, commonality and nuance, usually first for single cases (first interview) then subsequently across multiple cases (next interviews). Then, an overview was taken to allow for the development of a 'dialogue' between the researchers, their coded data and their knowledge about what it might mean for participants to have these concerns in this context, which moved the analysis towards a more interpretative account. The next step was to develop a structure/frame or what Smith et al. [99] call a gestalt which showed the relationship between the themes. Throughout the process, the material was organized in a format which allowed for the tracing of analyzed data from initial utterances to final themes, so that in effect, an audit trail would be evident.

Reliability and Validity

To improve the validity of the research, member checks, or member validation [101,102] were carried out on the research transcripts with the participants. Following steps by Stitt and Reupert [100], once transcriptions were completed, the completed transcripts were sent to the participant to read through to identify any inconsistencies or incorrect information that was contained within. No participant requested modifications or exclusions from the completed transcripts.

To ensure accuracy in the development of themes, inter-rater reliability testing was carried out [103]. An additional researcher identified themes from a subset of the transcripts and a comparison was made. Agreement was made following discussions between researchers [100].

### 3. Results

The results from the qualitative analysis will now be described. For clarity purposes, the results are presented in line with the factors as outlined in the literature review in relation to the specific traits of authentic leaders. The authors also considered presenting the results in line with codes and themes which were most frequently identified, however, following discussion and deep engagement with the literature, this option was deemed to be inappropriate for several reasons. Firstly, frequency of responses is a quantitative approach that misaligns with the theoretical positioning of the authors and qualitative research in general. IPA, which is adopted in this study, relies on the richness of the extracts and the themes, not the frequency or prevalence at which they are discussed by participants [99,100,104]. Furthermore, with respect to the literature, there are numerous

papers which highlight problems that arise from incorporating numbers into qualitative research [105–109]. The inclusion of quantitative processes can lead researchers away from the insiders' perspectives by focusing on the assignment of objective characteristics [106]. By reporting the frequency at which a particular finding, theme, or code emerges, there is "a danger of reducing evidence to the amount of evidence" [107] (p. 480) and overlooking the qualitative nature of the evidence. For instance, the perception of one participant is still valued in qualitative research, but to assign quantitative measures would reduce the relevance and impact of this perception. Furthermore, the use of numbers in the representation of results when using a relatively small population, as is common with qualitative research, may also make results appear more generalizable to a wider population than is fair to suggest [107]. For example, "90% of the sample" may be assumed to be quite large in comparison to "9 out of 10 participants agreed". Finally, as previously discussed, previous literature has highlighted that the implementation of numbers and frequency measures lends itself to the objectivist paradigm which is incompatible with qualitative research's subjective and constructivist underpinnings. For those reasons, this approach was deemed wholly inappropriate.

### 3.1. Authentic Leadership Traits

Evidence of authentic leadership traits as discussed in the literature review section of this paper, was identified in the participants (Figure 1). Barriers and facilitators to the performance of authentic leadership, of which the importance of consideration was discussed in the literature review section of this paper, were also identified and will be discussed at the end of the chapter.

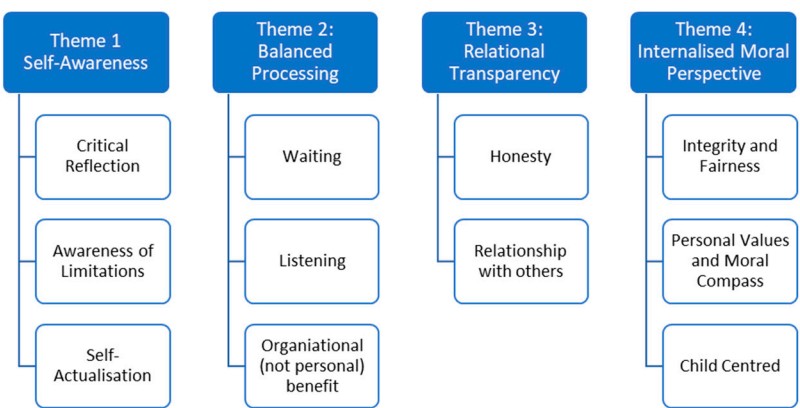

**Figure 1.** Authentic Leadership Traits.

### 3.1.1. Theme 1: Self-Awareness

As discussed in the literature review section of this paper, evidence of the self-awareness aspect of authentic leadership would manifest itself through self-reflection of one's personal values/morals, and assessing one's own competencies, strengths and weaknesses. Self-awareness emerged during discussions with participants in terms of critical reflection, awareness of limitations, self-actualization and transformation.

Critical Reflection

Evidence of critical reflection emerged as a dominant finding throughout the interview process with all participants acknowledging it as vital for successful leadership and highly integrated into their leadership styles. This was reflected in comments such as:

> A reflective practice paradigm has always been a part of the way that I work, the way that I lead. (P5)

> Being able to look back on decisions, question them, interrogate them, was the decision right? Did I take the right steps? Did I consult the right people, or did I just go, "this is a great idea! Bang!". (P7)

This element of critical reflection was not only evident in terms of self-reflection, but also in accessing external perspectives:

> Sometimes it is difficult to facilitate [critical reflection] for yourself. We are always asking "can we do it better?" You often need that outside perspective to really make a change. (P3)

> We all need a "critical friend," someone who can tell us if we are about to make that thunderous mistake or tell us to think about this decision or that decision. It's vital because sometimes you can be so focused on the goal, you miss something. (P4)

Despite the importance of critical reflection, participants identified difficulties such as an inability to receive authentic answers from others:

> I can find it difficult to get authentic answers because I think a lot of them are afraid of upsetting me if they are critical. (P7)

A possible negative outcome of critical reflection was the possibility of being overly critical of oneself:

> You have to be very careful not to bash yourself too much in that, in that critical reflection. (P6)

Awareness of Limitations

The majority of participants identified the leader's capacity to identify their limitations as being as important, if not more important, than identifying their strengths.

> I would have particular strengths, but I also have limitations . . . it is important that you recognize those limitations and embrace people who fill those gaps for you. (P3)

> You have to consider and evaluate your own role. I don't think any of us are very good at seeing our own flaws as openly as others, the first thing I try to do is see how this looks, outside of me. (P8)

In line with an awareness of limitations emerged the importance of acknowledging one's mistakes:

> I have to be willing to acknowledge that I have made mistakes . . . that is really important. (P2)

> I would hope I wouldn't make the same mistake twice. I'd make a different one, but I would genuinely try to not make the same mistake twice. (P4)

Self-Actualization

Although not as frequently as the other subthemes of self-awareness, self-actualization was identified by one participant as being important in authentic leadership:

> To be in control of and able to recognize your emotions, that is a huge part of being authentic. You are constantly trying to self-actualize, and you can have a huge influence if you are genuine. (P4)

3.1.2. Theme 2: Balanced Processing

As discussed in the literature review section of this paper, balanced processing will be exhibited through leaders' efforts to seek out the opinions of others, an openness to hear opposing views and time for self-reflection of the information received to allow for balanced decision making. Balanced processing emerged in the interviews with reference to waiting, listening and consideration of organizational (not personal) benefits.

Waiting

In order to allow for balanced processing, participants discussed the importance of waiting. This was to facilitate reflection and time to process the information:

> I quite often find myself time and say, "look, I'll get back to you tomorrow," or "can you leave that with me?" ... And then reflect and plan for making that decision. You have to wait, seek advice, and then respond. (P2)

> In conflict situations, one person always has to remain thinking. You need to buy yourself some time and think about how best to handle the situation and reach a resolution. (P3)

#### Listening

All participants discussed the pivotal role of listening when making balanced decisions.

> Listening, listening is vital. You need to listen to the staff, the parents, listen to the children. But then you need to make the decision. Examine your actions, what is the strength of this decision, what is its weakness, who will it impact? If you have listed to all parties, then you can make the correct decision and stand over it. (P2)

> People need to feel that they are heard ... ask people for their feedback. Take time and wait. So, listen, take it in, reflect upon it, and then move on to whatever needs to be done. (P1)

#### Organizational (Not Personal) Benefit

Participants in this study highlighted the importance of an "organization first" motivation in their balanced processing. This guided participants' decision making in ensuring they were considering the best decisions for the school and its community and not the leaders personally.

> You are doing it for the betterment of your school, not yourself, you must have the self-awareness to realize it is not about you. (P2)

> It's not about me, this operation isn't about me, it's about the children, making it better for the children. (P3)

#### 3.1.3. Theme 3: Relational Transparency

As discussed in the literature review section of this paper, relational transparency will be demonstrated through leaders' representation of themselves honestly, through clear and direct communication of their personal values and goals. Relational transparency was exhibited by the participants in this research in terms of honesty and a desire to promote relationships with their followers.

#### Honesty

Relational transparency was identified in participants through their willingness to be honest with others. For example, "openness" was identified by one participant as being at the "foundation of everything you do" (P1). Furthermore, honesty was important when the school leader had made a mistake:

> You are not going to work for 35 or 40 years and never make a mistake. In my experience, even when I have made a mistake is "hands up" ... don't obfuscate others to try and cover it. Be honest. (P6)

#### Relationship with Others

Participants in this study identified the importance of strong, positive relationships between the leader and others within the school. Firstly, collaboration emerged as important and was only made possible by leader buy-in:

> Collaboration is so important when we consider the day-to-day running of our schools. I am a firm advocate of mentoring ... .it has helped me, and I think that the whole idea of observation and supporting teaching and working together ... .it is becoming sort of a school philosophy. (P1)

Secondly, participants identified the efforts they employed to prevent or escalate conflict and maintain positive relationships:

Sometimes we can catastrophize situations and think 'Oh my God this is going to be an abject disaster'. Often times, those situations lead to something of nothing. More often than not it's the situations that you don't anticipate as being problematic are the ones that explode. (P5)

You need to take the time to plan what you are going to do and say if you can at all. Be proactive in trying to deal with situations meet them at the door rather than letting it get to a place where you are on the back foot and scrambling. (P2)

3.1.4. Theme 4: Internalized Moral Perspective

As discussed in the literature review section of this paper, authentic leaders will have a good understanding of their personal values and morals and this will guide their decision-making behaviors. The internalized moral perspective of participants strongly emerged in the analysis with subthemes identified in relation to integrity and fairness, personal values and moral compass and child-centered motivations.

Integrity and Fairness

Participants in this study discussed how decisions may not always be popular but need to be made based on fairness and integrity. This was exemplified by P3 and P9:

You would like to think that you always act fairly, don't you? Ultimately, if you treat people fairly, all that they want is respect. You often have to manage expectations when it comes to general opinions of "fairness." When it comes down to it, you have to trust that you are doing the right thing and making the right judgement calls. (P3)

I think any decision I take now; I take with the view that across the table from me is somebody from the Department of Education, somebody from the Board of Management, somebody from the Union, a parent. So whatever decision you do make, you can stand over to all the parties. You're expected to be, and I'll use the phrase that I learned and going back a few years ago, 'whiter than white'. Your decisions have to be transparent and made with integrity. (P9)

Personal Values and Moral Compass

Participants in this study identified their own moral compass and personal values as influencing their decision making in their role as school leaders. Participants identified their core values as stable over time as indicated by P3 when they said "Over the last thirty-odd years, my core values haven't changed." This was further supported by P6 in the following

At times, this led to extreme moral and personal turmoil as highlighted by. (P6)

You want to help … but then you need to step back into a space and ask yourself the question, "does this child's right to be educated in a school of their parents choosing, trump the rights of every other child in the class to be safe" It is the toughest decision to take. Morally you are torn. (P6)

Child Centered

The focus of the child at the center of the decision-making processes was evident across all participants. This was highlighted by P6 when the said "If it doesn't benefit the children, then we are doing something wrong." This was additionally supported by P3 in the following:

Ours is the business of education, so, for me the child is central, the child is the client. Everything you do is based on children and progressing children in

their experience of education. Every action that you take in your leadership is predicated on progressing the child. (P3)

*3.2. Barriers/Facilitators to Authentic Leadership*

As discussed in the literature review of this paper, it is necessary to explore the barriers to and facilitators of authentic leadership from the individuals' perspectives. Barriers to and facilitators of authentic leadership emerged in this study following the probing questions as outlined in the methods section of this paper, predominantly in relation to policy and procedures and school culture.

3.2.1. Policy and Procedures

The increase in mandated success criteria placed on school leaders by external stakeholders such as the DES, was identified as a barrier to school leaders' ability to lead authentically and in alignment with their own values and morals. In the first instance, policy constricts the freedom of school leaders to make decisions.

> The decisions we make have to be informed by the kind of wider landscape that we operate in. We don't operate in a vacuum and distinct from department policy, or maybe the patronage of the school we work in or the community that we work in. (P8)

In the second instance, policies were identified as at times conflicting with the school leaders' personal values and morals, resulting in a misalignment.

> Schools that focus on relational awareness tend to have better outcomes than schools that focus purely on testing and standardization. Teaching the skills of how to co-exist and work with others is key to enhancing learning. (P10)

The data-driven nature of schools following PISA results was identified by participants as opposing their moral values. This was exemplified by one principal who stated they were feeling "morally twisted and contorted" (P8), due to the direct negative effect these results were having on students, particularly special education students:

> The wider system is only interested in what is measurable. The contribution then of those children with special education needs becomes less, so, as far as the system is concerned, because their contribution isn't measurable. That to me is reprehensible. (P8)

Despite participants valuing a holistic, child-centered approach to leading, this was not always in alignment with the data-driven nature of the system making it difficult to lead based on these values, as highlighted in the following:

> We're looking at what's under the microscope, and you know, that wider piece about the social and emotional and spiritual development of children in schools has definitely been side-lined. And you know yourself and myself, as teachers, as principals in schools, we do try and defend it. You know, it's very important to us, but I mean, it's becoming more and more difficult to defend the prominence of that in school when the wider system is only interested in what is measurable. (P8)

3.2.2. School Culture and Context

The culture and context of a school were identified by participants as having the potential to act as either as a barrier to, or facilitator of, authentic leadership. The swing of the proverbial pendulum was attributed to the values held by the school itself and how they aligned with that of the leader. For example, during alignment, being authentic is more easily achieved:

> There isn't a conflict between my values and the ethos of the school. There is an alliance between them and the organization I am leading, I couldn't be true to myself and do my job if they didn't. (P5)

Furthermore, learning the values of the school can take time, and this may also influence one's ability to be authentic. This was indicated by P3 when they said "The style of leadership you adopt depends on the context you arrive into" and P2 when they said "You values can often depend on your context." This was further elaborated by P2 in thefollowing:

> I really value the opinions of others I suppose in my present role I entered a school that I was not part of the staff in, so I needed to learn the culture of the school. Listening to the staff and trying to figure out who is genuine in their actions and who has hidden agendas. (P2)

Although many of the leaders in this study identified an alignment between their values and that of the school they worked in, which could be a reflection of the long tenure many of the participants had within the schools they worked in, conflict could arise at times between the values of the child-centered approach at the leaders' core, and the values system of the school:

> You are trying to protect the baseline value system but also trying to open it up to be inclusive and supportive of the child who says they are gay or bisexual. It can be a challenge. (P3)

## 4. Discussion

The results of the data analysis allowed for the identification of authentic leadership traits in all of the participants. However, although possessing the personal traits that allow one to be an authentic leader, the ability to enact authentic leadership can be hindered or facilitated based on the school culture and power dynamics within the structure of the educational system. These findings will now be discussed in further detail.

Consistent with Ferrero et al.'s [35] conceptualization of authentic leadership, school leaders described self-awareness, relational transparency, balanced processing and internalized moral perspective. In terms of self-awareness, Irish primary school leaders interviewed in this study took time to critically reflect upon themselves and their decisions as leaders and indicated an awareness of their limitations. In relation to balanced processing, the school leaders in this study indicated a preference for listening to a variety of opinions from multiple stakeholders, taking time to consider options and considering information before making any decisions. Participants identified being honest with others in line with relational transparency. Finally, participants' leadership decisions were shown to be informed by their personal morals and values, which is core to authentic leadership [28,35]. Although previous literature has suggested that the morals at the center of the authentic leader may be amoral or unethical from a normative standpoint [33], the leaders in this study indicated values pertaining to the holistic, child-centered focus which would be in line with values typically accepted as moral/ethical [110].

Given the potential benefits of authentic leadership for teachers, such as reduced turnover intentions [68,74] and increased psychological empowerment [68], this finding is encouraging as it identifies authentic leadership traits in school leaders, and in so doing, highlights the possibility of these traits to be fostered and promoted in school leaders, with the potential of tackling the current staffing crisis being experienced in Ireland [75] and beyond [79,80]. However, despite the identification of these traits, before there is any consideration of how they may be fostered, it is first imperative to acknowledge the barriers preventing school leaders from being authentic leaders. Findings from this study show that the ability of school leaders to adopt an authentic style of leadership is directly impacted by the culture of the school they worked in. Participants in this study highlighted that when their personal values and morals were in alignment with the culture of the schools they worked in, the ability to lead authentically was more easily facilitated than when they did not. It is important to note that all participants identified some alignment between their own values and that of the school they worked in, which is possibly a reflection of the long tenure they had within their respective schools. This is supported by the findings that new teachers' values were identified as misaligning more so than that of more tenured teachers.

However, this is only a suggested cause and future research with a larger participant sample would go towards investigating any causal relationships that may exist.

Although acknowledging times of alignment, misalignment was still identified at times, and upon its occurrence, participants discussed the need to lead based on the school's needs in order to have any form of influence. This was particularly true when taking up new positions in a school one had not yet become familiar with. This finding is consistent with previous literature that has suggested the organization is a prerequisite for not only the adoption of leadership styles in general [69] but authentic leadership specifically [68]. This finding also expands the field of knowledge by showing that organizational cultures in schools, and not only in the field of business, can influence the adoption of an authentic leadership style. In moments of misalignment, it is clear that one's freedom to be an authentic leader is challenged. In order to be successful, this suggests the need for leaders to clearly communicate their values and morals to the wider organization to foster alignment [111]. For those leaders drawn towards authentic leadership, this may be more easily fostered due to their tendencies for relational transparency and openness to feedback from others [35]. However, there needs to also be a consideration of power dynamics in terms of whose values are valued, and whose values are listened to [112] as this may directly impact a person's freedom to lead authentically, particularly those from marginalized groups [113]. This is an area that warrants further investigation.

As evidenced by the findings of this research, authentic leadership may also be hindered by misalignment between leader values and policies and procedures. External markers of success have placed pressures upon leadership styles that value child-centered practices instead resulting in a requirement for performativity and the achievement of success criteria. Worryingly this was identified as being particularly detrimental for schools with special educational needs. This supports previous research which acknowledges that the impact of accountability and emphasis on performance measures may directly impact the leadership styles of school leaders [114]. It certainly warrants further investigation. Although there have been advancements in recent years in affective education and the importance of consideration of the student as a "whole person," the influence of performativity and metric-driven education may obscure the primacy of child-centered approaches that lie at the core of authentic leaders' values. Criticisms of external markers of success in education are neither new nor limited [115–117], but the findings from this research add an additional lens to the available literature by highlighting that this agenda may also be adversely impacting leaders' desires to lead authentically and, consequently, limit the associated benefits for all.

*Research Strengths and Limitations*

The present study examined the existence of authentic leadership traits in Irish primary school leaders. The scoping nature of the study and the qualitative design allowed for the direct exploration of leaders' perspectives and experiences rather than the more common follower perspectives which is problematic [86] when seeking to understand leadership more deeply. Although this is a strength of the paper, it is acknowledged that the self-reporting nature of the interview process also poses threats such as social desirability bias [118]. Authors undertook specific actions to limit this possibility by use of open-ended, un-leading questions, however, future studies may benefit from additional assurances such as the incorporation of a paired leader–teacher design, as asking for others' perspectives may reduce errors due to social desirability bias [118].

The sample size of ten participants is a recognized limitation of this study. Convenience sampling, as was adopted in this study, often results in small numbers of participants in similar research designs [99,119]. However, this is an initial scoping study to ascertain if this area warrants further investigation. The authors do not seek to generalize from this study. Indeed, there are arguments to be made against the generalizability of findings [120] and quantitative studies have been identified as possibly complementary methods for

addressing these concerns [86]. Therefore, future studies may benefit from a mixed-method approach to data collection.

## 5. Conclusions

Authentic leadership has been identified as the "golden standard" of leadership [20] (p. 54), yet, as discussed in the literature review section of this paper, is relatively understudied in Irish primary school leaders. This paper provides perceptions from Irish primary school leaders in relation to their experiences of leadership and an identification of authentic leadership traits in a school system where distributed leadership theory, policy and practice is dominant, therefore offering evidence that alternative leadership theories can be prevalent.

This is, to the authors' knowledge, one of the first, if not the first, research piece to explore this topic within this unique subset. Considering the prevalence and dominance of distributed leadership within not only Irish primary schools [48], but in school policies and systems across the globe [49], and the associations between distributed leadership and burdening of employees [59] as a means to satisfy the neoliberal agenda [48], the current research presents an alternative consideration. As discussed in the literature review, this is not to say that authentic leadership should replace the distributed leadership system but may in fact strengthen it by supporting its successful implementation and empowerment potential.

Considering the potential benefits of authentic leadership, the results of this study support theory and research suggesting that authentic leadership may be hindered or facilitated by organizational culture and external policies. This is important for consideration in educational practice as the potential benefits of authentic leadership, including as a buffer against the negative associations of distributed leadership, should be encouraged and not stifled. In one such example as drawn from this research, leaders are not always able to lead based on their values of holistic education, and instead need to facilitate the neoliberal agenda for performativity and grade attainment. In line with this, future policymakers should consider open discussion with school leaders around the prevalence of the neoliberal agenda and its impact on the holistic side of student engagement which aligns with, at the very least, a subset of the leader population, as evidenced in this research.

Apart from practice, the results of this study add to our knowledge of leadership theory both in pedagogy and beyond. This study addresses calls for the study of authentic leadership to be conducted from the leader's perspective and highlights the approach as having something meaningful to offer for this purpose. This research also highlights some additional concerns for the theory of authentic leadership research, such as the freedoms available to adopt an authentic leadership style in the workplace. In doing so, the data suggest the need for further studies in this field.

**Author Contributions:** All authors listed in this document contributed significantly to the research discussed within the paper and authorship of the paper itself. All contributing authors and personnel have been included as authors in this paper. Conceptualization, J.L., D.D. and P.M.M.; Methodology, J.L., D.D. and P.M.M.; Formal analysis, J.L.; Investigation, J.L.; Visualization, N.L.; Writing—original draft, N.L.; Writing—review & editing, J.L., D.D., N.L. and P.M.M.; Visualization: N.L.; Supervision, D.D. All authors have read and agreed to the published version of the manuscript.

**Funding:** This research received no external funding.

**Institutional Review Board Statement:** The research discussed in this paper received ethical approval from the Education and Health Sciences Ethics Department, University of Limerick. Approval number: EHSREC no. 2020_03_36_EHS.

**Informed Consent Statement:** Informed consent was obtained from all participants in this study.

**Data Availability Statement:** Data available on request due to restrictions placed on participant privacy, however, segments of the data can be requested by contacting the corresponding author.

**Conflicts of Interest:** The authors declare no conflict of interest.

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
