# Peer review of "The Real Deal: A Qualitative Investigation of Authentic Leadership in Irish Primary School Leaders"

_societies, doi:10.3390/soc12040106_

Round 1

Reviewer 1 Report

The study of the importance of authentic leadership is considered relevant and interest and is a qualitative study that includes the perspective of those involved, although it needs improvement. It would be important for the author to strengthen his theoretical framework in a way that demonstrates the innovative contribution to the theory and concepts he presents. In addition, the author should review and reword the objectives, improve the wording of the method and the presentation of results based on the specific objectives of the research. It should avoid simple conceptual description and seek to connect the fundamental concepts and objectives of the paper by improving the discussion and conclusions. In addition, it should also explore the concrete implications of the conclusions reached, for example, what actions could be taken to educational practice. There are a number of arguments in favor of its publication and others that, being unfavorable, can be corrected by the author. If the author corrected the problems indicated in the weaknesses of the article, quality standards could be achieved in said article that would make it worthy of acceptance. Therefore, my decision is: Favorable to its publication on the condition that the difficulties that have been detailed are removed. See attached document for additional comments to Author

Author Response

Dear reviewer,

Thanks for your comments and suggestion, authors have made the revision according to your review report and you can check details according to the attached file.

Thanks

Reviewer 2 Report

Overall Impressions

I have some reservations whether this study significantly adds value to the field. The research evidence here is relatively weak, which resulted in a dispersed and thin set of findings. I think a manuscript describing the current state of research on authentic leadership would make a more valuable contribution to the literature. I’m less concerned about the size of the sample and more with the depth of analysis. To a degree, it’s also not fully clear what the paper is trying to accomplish. The authors are highlighting “authentic leadership,” an idea they admit is not consistently defined, and make good attempts to define it overall. However, they avoid contrasting authentic leadership with other forms of leadership in a way that might better inform their premise. In addition, the manuscript seems to focus on leaders' morals, and the difference in outcomes from aligned vs mismatched morals, but the authors do not satisfactorily address what they mean by morals. It is left ambiguous as to whether they are referring to social morals or, say, organizational values.

Specific Comments

Intro

Page 1, 1st paragraph: Consider citing Grissom et al. Wallace synthesis on leadership effects on student achievement. Grissom, Jason A., Anna J. Egalite, and Constance A. Lindsay. 2021. “How Principals Affect Students and Schools: A Systematic Synthesis of Two Decades of Research.” New York: The Wallace Foundation.  

Para right below Table 1 is helpful, but falls short of offering a clear description of the four constructs. Perhaps spend more time unpacking these 4 dimensions of AL.

I question whether “internalized moral perspective” is a useful construct in this context. Whose morals or ethics? Russia’s Putin likely follows his own values with conviction…is he an authentic leader?

To move on to antecedents of authentic leadership seems premature. Has the construct of authentic leadership been successfully measured? validated?

The manuscript reads: “What this appears to suggest is an intersection between the personal and the organizational dimensions in the development a leadership styles, a clear departure from traditional understandings of the charismatic and hero leader.” I like this notion and it advances thinking on this topic.

I believe on p. 2, this statement is not clear to me: “The concept of authentic leadership suggests that authentic leaders’ actions are based on their own moral and ethical beliefs [18, 27] and in doing so, promote trust, authenticity, well-being, and sustained performance in their followers [25, 28].”

Methods

I’m not convinced that purposive sampling was used here. Purposive is what regard? It seemed to be more of a convenient sample given that a population of primary school leaders were asked to volunteer to participate.

More information on the types of questions posed in the interview protocol would be helpful, even if a couple of examples. I don’t have a sense, as a reader, if questions aligned with the AL framework or if questions were more general in terms of how these leaders perceived leadership. When I read some findings, I wonder why leaders referenced things like “critical reflection” without prompts. I think it’s important for the reader to know.

I appreciate the thorough description of IPA, but what does this phrase mean: “close consultation with supervision”?

Looking for positionally or subjectivity or trustworthiness section.. wasn’t in methods.

Findings

The IPA analysis seems to suggest, as most interpretative approaches, an inductive, emergent analysis. But the findings immediately are presented in relation to Ferraro’s construct of authenticity. Kind of suggests, at worst, confirmatory bias or, at best, a lack of deep inductive analysis. I can see fitting the findings (themes) to juxtapose against the AL framework, but later on – not as a fundamental part of the analysis.

Figure 1 also references Ferrero et al.’s four AL traits, but the figure itself includes themes that appeared to derive from the study. Or perhaps I’m confused and the sub-themes also are part of Ferrero’s construct.

The findings offered, in some cases, illustrative quotes for each sub-theme under each of the four AL traits. However, I found the process a bit reductionistic. The result is a sparse write up for each theme and sub-theme and the analysis is spread too thin. The research warrants for each sub-theme thus appear weak in many places or non-transparent. The authors report multiple themes in line with their view of authentic leadership, and present quotes from interviews to support this, but do not provide general density of themes or ideas to a point that sufficiently lends itself to their claims about the relative strength/ prominence of each theme.

Under the heading: 3.2.1.3. Self-actualisation

“Although not as frequently as the other subthemes of self-awareness, self-actualisation was identified by one participant as being important in authentic leadership”    I question why one participant quote is worthy of reporting as an overall theme? Certainly variation in responses among participants is important and every finding does not have to be consensus across participants. However, the way it’s written it feels as important as a participant-wide theme.

I think the Limitation section should emphasize the fact that the study relies on the self-reports of participants. Both a strength — to access perspectives of leaders — but also a clear limitation in terms of research warrant.

Discussion

Overall, I think the Discussion section was the stronger part of the paper and had the most value along with the intro sections and lit review. However, the discussion could occur almost without the empirical data. It suggests a conceptual paper would be stronger, because the empirical part here is lacking. I’m less concerned about the small sample and far more with the analysis and findings write up.

Lastly, I do not know if the start to the conclusion is fully reasonable to include: “Authentic leadership has been identified as the “golden standard” of leadership yet is relatively understudied in primary school leaders.”

Author Response

(The authors gave the same response as above.)

Reviewer 3 Report

The study was conducted appropriately and the manuscript is well-written

Author Response

Dear reviewer,

Thanks for your review and your positive feedback.